# DNA Deamination Is Required for Human APOBEC3A-Driven Hepatocellular Carcinoma In Vivo

**DOI:** 10.3390/ijms24119305

**Published:** 2023-05-26

**Authors:** Jordan A. Naumann, Prokopios P. Argyris, Michael A. Carpenter, Harshita B. Gupta, Yanjun Chen, Nuri A. Temiz, Yufan Zhou, Cameron Durfee, Joshua Proehl, Brenda L. Koniar, Silvestro G. Conticello, David A. Largaespada, William L. Brown, Hideki Aihara, Rachel I. Vogel, Reuben S. Harris

**Affiliations:** 1Department of Biochemistry, Molecular Biology and Biophysics, University of Minnesota, Minneapolis, MN 55455, USA; jnaumann@umn.edu (J.A.N.); argyris.2@osu.edu (P.P.A.); brown344@umn.edu (W.L.B.); aihar001@umn.edu (H.A.); 2Masonic Cancer Center, University of Minnesota, Minneapolis, MN 55455, USA; temizna@umn.edu (N.A.T.); konia001@umn.edu (B.L.K.); larga002@umn.edu (D.A.L.); isak0023@umn.edu (R.I.V.); 3Howard Hughes Medical Institute, University of Minnesota, Minneapolis, MN 55455, USA; 4Division of Oral and Maxillofacial Pathology, College of Dentistry, Ohio State University, Columbus, OH 43210, USA; 5Department of Biochemistry and Structural Biology, University of Texas Health San Antonio, San Antonio, TX 78229, USA; carpenterm2@uthscsa.edu (M.A.C.); guptah@uthscsa.edu (H.B.G.); cheny20@uthscsa.edu (Y.C.); zhouy4@uthscsa.edu (Y.Z.); durfeec@livemail.uthscsa.edu (C.D.); proehl@uthscsa.edu (J.P.); 6Howard Hughes Medical Institute, University of Texas Health San Antonio, San Antonio, TX 78229, USA; 7Institute for Health Informatics, University of Minnesota, Minneapolis, MN 55455, USA; 8Core Research Laboratory, ISPRO, 50139 Florence, Italy; s.conticello@ispro.toscana.it; 9Institute of Clinical Physiology, National Research Council, 56124 Pisa, Italy; 10Department of Pediatrics, University of Minnesota, Minneapolis, MN 55455, USA; 11Department of Obstetrics, Gynecology, and Women’s Health, University of Minnesota, Minneapolis, MN 55455, USA

**Keywords:** APOBEC3A, carcinogenesis, DNA deamination, DNA mutation, hepatocellular carcinoma, molecular mechanism, RNA editing, tumorigenesis

## Abstract

Although the APOBEC3 family of single-stranded DNA cytosine deaminases is well-known for its antiviral factors, these enzymes are rapidly gaining attention as prominent sources of mutation in cancer. APOBEC3′s signature single-base substitutions, C-to-T and C-to-G in TCA and TCT motifs, are evident in over 70% of human malignancies and dominate the mutational landscape of numerous individual tumors. Recent murine studies have established cause-and-effect relationships, with both human APOBEC3A and APOBEC3B proving capable of promoting tumor formation in vivo. Here, we investigate the molecular mechanism of APOBEC3A-driven tumor development using the murine *Fah* liver complementation and regeneration system. First, we show that APOBEC3A alone is capable of driving tumor development (without *Tp53* knockdown as utilized in prior studies). Second, we show that the catalytic glutamic acid residue of APOBEC3A (E72) is required for tumor formation. Third, we show that an APOBEC3A separation-of-function mutant with compromised DNA deamination activity and wildtype RNA-editing activity is defective in promoting tumor formation. Collectively, these results demonstrate that APOBEC3A is a “master driver” that fuels tumor formation through a DNA deamination-dependent mechanism.

## 1. Introduction

Human cancers typically manifest at least one mutational mechanism that drives genomic instability and tumorigenesis. Well-established examples include mutagens such as cigarette smoke and ultraviolet (UV) light, as well as genetic factors such as deficiencies in DNA mismatch repair, recombination repair, and DNA polymerase proofreading (e.g., *MSH2*, *BRCA1*, and *POLE*, respectively). A more recent example is the APOBEC3 family of single-stranded DNA cytosine-to-uracil (C-to-U) deaminases that is strongly implicated in causing C-to-T and C-to-G mutagenesis in TCA and TCT motifs in many different cancers (COSMIC single base substitution (SBS) signature 2 and 13, respectively) [1,2,3,4]. Simple C-to-U deamination events can also lead to more toxic lesions, including abasic sites and DNA breaks, that together precipitate replication stress and larger-scale genomic aberrations, including chromosomal instability (CIN) [5,6,7].

Although the human APOBEC3 family comprises seven normally antiviral enzymes, only two—APOBEC3A (A3A) and APOBEC3B (A3B)—have emerged as leading culprits in cancer [1,8,9,10]. Both enzymes preferentially catalyze C-to-U deamination in the trinucleotide motifs described above and are clearly capable of accessing chromosomal DNA as evidenced by triggering strong DNA damage responses upon overexpression in human cells [8,11,12]. In addition, A3A and A3B mRNA expression levels have been associated with mutation loads and clinical outcomes in multiple tumor types [8,13,14,15,16,17]. Overall, a model is emerging in which A3A/A3B mutagenesis is responsible for a significant amount of molecular and phenotypic heterogeneity in a wide variety of different human cancers [10,18,19].

However, an unfortunate evolutionary impediment to studying this process is that rodents, including laboratory mice, encode only a single APOBEC3 enzyme, which is a weak deaminase, is cytoplasmic, and is unlikely to damage genomic DNA [20,21,22]. Murine tumors in a range of model systems are accordingly homogeneous and rarely reflect the heterogeneity of human disease. Therefore, considerable effort has been invested to address this issue by developing murine models for mutagenesis and carcinogenesis by human A3A and A3B [23,24,25,26]. Thus far, the *Fah* hepatocellular carcinoma system is the only one to enable the reconstitution of human cancer mutation signatures SBS2 and SBS13 in murine tumors [24]. In brief, *fumarylacetoacetate hydrolase* (*Fah*)-mutant animals are injected through the tail vein with a plasmid DNA construct encoding wildtype Fah and human A3A, and within 48–72 h, approximately 5–10% of hepatocytes show positivity for A3A protein expression by immunohistochemistry (IHC). Due to transgenic expression of the *Sleeping Beauty* transposase, SB11, most of these hepatocytes incorporate the *Fah*/*A3A* expression cassette heritably into the genome by cut-and-paste transposition. Concomitant withdrawal of NTBC (2-(2-nitro-4-trifluoromethylbenzoyl)-1,3-cyclohexanedione) from drinking water imposes a selection for cells that stably express wildtype *Fah* and are thereby able to avoid the toxic accumulation of tyrosine catabolites [27,28,29,30]. In our original studies, 6 months after A3A construct hydrodynamic injection and NTBC withdrawal, a five-fold increase in the development of hepatocellular carcinoma is evident [24].

Here, we utilize the *Fah* liver regeneration system to explore the molecular mechanism of human A3A-driven carcinogenesis in vivo. First, we show that A3A alone can drive hepatocellular carcinoma formation. Our prior work included a knockdown construct for *Tp53* [24], and here, we show that this additional manipulation is dispensable for the observed tumor phenotypes. Second, we demonstrate that the catalytic activity of A3A is required for tumorigenesis, as an otherwise isogenic E72A mutant construct fails to promote tumor formation. Third, we track the kinetics of tumor formation and show that micro- and macro-sized lesions can form in as little as two months. Last, we use a unique separation-of-function mutant to show that A3A DNA-deamination (and not RNA-editing) activity is required for tumor formation in vivo. Taken together, these studies support a model in which genomic DNA deamination by wildtype human A3A causes a sufficient level of DNA damage and mutation to trigger the initiation and progression of hepatocellular cancer.

## 2. Results

### 2.1. APOBEC3A Alone Is Sufficient to Promote Tumorigenesis

Our prior studies demonstrated increased frequencies of hepatocellular carcinoma in a *Fah* liver regeneration model 6 months after simultaneous knockdown of murine *p53* and expression of human A3A from a *Sleeping Beauty* transponson construct (SB11) [24]. The rationale for this combination of perturbations was based in part on earlier work using the *Fah* system showing that p53 knockdown accelerates the development of hepatocellular hyperplasia [29]. The rationale was also based on earlier findings in cancers, such as breast cancer and head and neck squamous cell carcinoma, associating genetic inactivation of p53 and APOBEC3 mutagenesis [1,4,8]. One explanation for this association is that p53 inactivation may be required for cells to tolerate APOBEC3-catalyzed DNA uracilation and the ensuing DNA damage responses and thus can accumulate mutations without undergoing DNA damage-induced apoptosis.

We therefore first asked whether *p53* knockdown is mechanistically required for A3A-induced tumor formation. This was performed by injecting the tail veins of 2-month-old *Fah*-null SB11 animals with constructs expressing Fah and A3A or Fah alone, together with a separate construct for *p53* knockdown or a non-shRNA control (construct schematics in Appendix A). Immediately, post-injection (p.i.) animals were provided normal drinking water (non-supplemented with NTBC) and aged an additional 6 months to force liver regeneration and to allow time for possible mutation accumulation and tumor formation (Figure 1A). At 8 months of age, all animals were euthanized, and livers harvested for analysis. As reported [24], the *p53* knockdown condition yields a small number of visible tumors in a minority of animals (mean 0.4, median 0, range 0–2), whereas *p53* knockdown combined with A3A expression induces multiple, macroscopically detectable tumors in the majority of animals (mean 2.1, median 2, range 0–6; *p* = 0.013 when compared to control by Kruskal–Wallis non-parametric test with Dunn’s adjustment for multiple comparisons; Figure 1B,C). In comparison, control conditions yield no visible tumors (mean 0, median 0, range 0), and A3A expression alone induces significantly high levels of liver tumor burden (mean 3.2, median 2, range 0–9; *p* < 0.001 by Kruskal–Wallis non-parametric test with Dunn’s adjustment for multiple comparisons; Figure 1B,C). As an additional control, *p53* knockdown was confirmed by RTqPCR (*p* values calculated by Brown–Forsythe and Welch ANOVA with Dunnett’s adjustment for multiple comparisons; Figure 1D).

Histopathologic analysis was performed on representative formalin-fixed paraffin-embedded (FFPE) liver specimens from each experimental condition (Figure 1E). Fah-complemented control liver parenchyma exhibits normal cellular morphologic characteristics without any dysplastic changes (i.e., cellular and nuclear pleomorphism, atypia, nuclear hyperchromasia, increased mitotic activity). In contrast, the macroscopically evident A3A/shp53 liver tumors exhibit histopathologic features consistent with hepatocellular carcinoma (HCC) that comprise overt nuclear pleomorphism and atypia, increased cellular and nuclear size with irregular chromatin distribution and presence of macronucleoli, as well as numerous atypical mitotic figures (>3–4 per high-power field). Although only two mice in the *p53* knockdown cohort developed tumors, the majority of liver tissues in this group exhibit areas of moderate dysplastic alteration. Similar to A3A/shp53 livers, A3A liver lesions are invariably malignant and display high-grade microscopic characteristics as described above. Notably, in addition to readily detectable HCC masses, multiple discrete clusters of malignant cells (microlesions) are evident within the parenchyma of A3A livers, consistent with A3A induction leading to the development of HCC. A3A expression was confirmed at the protein level by IHC, although a markedly heterogeneous staining pattern is observed. Specifically, cell-wide, i.e., nuclear and cytoplasmic, expression of A3A is seen in normal adjacent-to-tumor (NAT) hepatocytes in A3A liver specimens, whereas HCC lesions are uniformly negative, most likely reflecting loss of A3A due to the selective pressure imposed by this potent DNA deaminase (Figure 1E; discussed further below). Combined, these data demonstrate that *p53* depletion is dispensable for liver carcinogenesis by human A3A and, importantly, that A3A alone is capable of driving liver carcinogenesis in this murine model system.

### 2.2. A3A Catalytic Activity Is Essential for Tumor Formation

We next asked whether the increased frequency of tumor formation caused by A3A requires catalytic activity. This is an important question because it is formally possible that stable expression of a human DNA deaminase such as A3A in murine tissues might deregulate normal processes, including those responsible for controlling endogenous APOBEC family members such as murine Apobec1. We therefore compared the tumorigenic potential of wildtype A3A and a catalytic mutant derivative E72A (construct schematic in Appendix A). Glutamic acid 72 is required for initiating the deamination reaction by deprotonating water and for creating the hydroxide that attacks the C4 position of the cytosine ring [31]. Accordingly, extracts from cells expressing wildtype A3A show strong single-stranded DNA deamination activity, and those expressing the otherwise isogenic E72A construct have no detectable activity (Appendix A). A small number of A3A-expressing cells also exhibit evidence for nuclear g-H2AX staining indicative of elevated levels of DNA damage (Appendix A).

Using the timeline described above, mice were enrolled randomly into A3A- or E72A-expressing groups. Six months following hydrodynamic injection, animals in the A3A cohort show enhanced liver tumor formation (mean 5.3, median 2.0, range 1–17; Figure 2A,B). In contrast, the E72A group shows no macroscopically or histopathologically detectable lesions (mean 0, median 0, range 0; *p* < 0.001 by Mann–Whitney non-parametric test; Figure 2A,B). This highly significant result was not due to a corrupt plasmid or a DNA delivery failure because nearly all E72A-regenerated livers showed near-complete reconstitution, as evidenced by cell-wide moderate-to-strong A3A immunostaining in >85–90% of hepatocytes (Figure 2C, right panels).

An intriguing observation from our original studies [24], again evident here, is dramatic variability in the overall pattern of wildtype A3A staining in Fah-regenerated livers. In the A3A conditions, with or without p53 depletion, A3A protein expression is never observed in HCCs or cancerous lesions, and it is only detected with considerable variegation in surrounding NAT liver tissues (Figure 1E, right panels; Figure 2C, left panels). This extreme variability is entirely dependent on catalytic activity, as evidenced by the relative uniformity of E72A staining (Figure 2C, right panels). Taken together with evidence for elevated g-H2AX staining (Appendix A), these observations indicate that wildtype human A3A causes a high level of DNA damage that is selected against during liver regeneration, such that A3A itself is lost early in tumorigenesis (i.e., lost well in advance of visible, likely clonal, tumor formation).

### 2.3. Kinetics of A3A-Catalyzed Tumor Formation

Our studies demonstrate that human A3A induces a substantive liver tumor burden within 6 months (Figure 1 and Figure 2). Many of these tumors are large (>1 cm^3^) and, in a few instances, A3A-reconstituted mice died before the 6 month endpoint could be reached. We therefore sought to determine the kinetics of tumor formation by analyzing A3A- and E72A-injected animals at earlier timepoints. The first experiment compared livers procured 8 weeks post-injection (p.i.) and, remarkably, even livers from this early timepoint showed high tumor burdens (A3A vs. E72A, *p* < 0.001 by Mann–Whitney non-parametric test; Figure 3A,B). Furthermore, by both immunoblot and IHC analysis, wildtype A3A protein expression is already absent from tumor masses with the exception of rare A3A-expressing lesional cells and heterogeneous in NAT hepatocytes (i.e., A3A is already selected against, Figure 3C,D). In sharp contrast, E72A expression is retained in phenotypically normal liver tissues (Figure 3C,D).

Since macroscopic tumor formation occurs within 8 weeks, an additional experiment was conducted to compare tumor formation at earlier 2–6 weeks p.i. timepoints. A small mass was evident in one animal at the 4-week timepoint, and multiple tumors were apparent in several animals by 6 weeks p.i. (white arrows denote tumorous growths; *p* values calculated by Kruskal–Wallis non-parametric test with Dunn’s adjustment for multiple comparisons; Figure 3E,F). Soluble extracts from representative livers from A3A-injected animals showed variable protein expression by immunoblotting with initially low levels at 2 and 3 weeks p.i. (potentially due to relatively few cells expressing A3A), moderate expression levels at the 4- and 6-week timepoints during Fah selection-driven liver regeneration (Figure 3G), and finally very low levels again by 8 weeks p.i. (potentially due to selection against A3A, as observed in Figure 3C). Similar A3A expression trends were apparent by IHC analyses of liver tissues procured at the same time points p.i. (Appendix A). These multi-timepoint studies combined to indicate that de novo tumor formation can occur in a relatively short period (i.e., weeks instead of months as typically reported for this system).

### 2.4. A3A-Catalyzed Tumorigenesis Is Driven by DNA Editing

A3A is a potent single-stranded DNA deaminase, but it is also capable of editing RNA cytosines, particularly in loop regions of hairpin structures [32,33]. This raises the intriguing possibility that some (and potentially all) of the aforementioned tumor phenotypes may be due to RNA-editing events. To address this possibility, we tested a recently described A3A separation-of-function mutant, A3A-Y132G-G188A-R189A-L190A (GAAA) [34], that was engineered to lack DNA deaminase activity, yet to retain full RNA-editing activity (structural model with ssDNA in Figure 4A, construct schematic in Appendix A). Consistent with this prior report, the quadruple A3A mutant has approximately 10-fold lower single-stranded DNA deamination activity in comparison to wildtype A3A in soluble extracts from 293T cells (protein expression, ssDNA deaminase activity and quantification in Figure 4B). A similar deficiency in DNA deamination is evident in a real-time C-to-T base editing assay [35] in which eGFP positivity is restored efficiently by wildtype A3A but not by the quadruple GAAA mutant (*p* values calculated by Brown–Forsythe and Welch ANOVA with Dunnett’s adjustment of multiple comparisons; Figure 4C).

In contrast, the quadruple mutant, GAAA, retains wildtype-like RNA-editing activity, as evidenced by sequencing the hairpin region of the *SDHB* mRNA in 293T cells expressing wildtype A3A, quadruple GAAA mutant, and catalytic mutant E72A derivatives (Figure 4D,E). Wildtype A3A exhibits an average of 23 ± 3.8% editing and GAAA 30 ± 2.3% editing (*n* = 3; *p* values calculated by one-way ANOVA with Tukey’s adjustment; Figure 4D,E). In addition, similarly robust RNA-editing levels are evident for both wildtype A3A and GAAA in a real-time RNA-editing assay in which A3A-catalyzed deamination of a single RNA cytosine leads to functional inactivation of a luciferase reporter (*n* = 3; *p* values calculated by one-way ANOVA with Tukey’s adjustment; Figure 4F). These DNA- and RNA-editing results combine to demonstrate that GAAA is deficient in DNA- but not RNA-editing activity.

Finally, the GAAA separation-of-function mutant was tested in the Fah system in parallel with an empty vector (negative control) and wildtype A3A (positive control) (construct schematics in Appendix A). Expression of the quadruple GAAA mutant was confirmed in vivo by IHC in liver tissues 72 h p.i. (Appendix A). Interestingly, 6 months p.i., GAAA was found to trigger significantly lower levels of tumor formation in comparison to wildtype A3A (*p* = 0.016 by Kruskal–Wallis non-parametric test with Dunn’s adjustment for multiple comparisons; Figure 4G,H). The numbers of tumors present in individual GAAA reconstituted livers were variable, ranging from zero to six tumors per animal, which is consistent with this protein retaining a low level of ssDNA deaminase activity. As above, for wildtype A3A in multiple experiments, protein level expression of the GAAA quadruple mutant is no longer detectable in tumor tissues at 6 months p.i. by IHC. Given that RNA-editing levels in two different systems are normal for the GAAA construct and that DNA-editing levels are reduced, again in two different systems, these results combine to indicate that A3A-catalyzed DNA deamination activity (and not RNA editing) is responsible for the tumor phenotypes described here using the Fah HCC model system.

## 3. Discussion

We previously showed that hydrodynamic injection of transposable constructs encoding human A3A and *shP53* are able to trigger HCC development in Fah-deficient, SB11-expressing mice [24]. Here, we demonstrate that A3A alone is capable of triggering similarly robust tumor phenotypes 6 months p.i. and, remarkably, that tumor formation becomes evident as early as 6–8 weeks p.i. A3A-driven tumorigenesis is completely dependent on functionality of the catalytic glutamate residue (E72), which is essential for deprotonation of water and hydrolytic attack of target cytosine nucleobases. Importantly, additional experiments with a separation-of-function quadruple mutant (A3A-Y132G-G188A-R189A-L190A; GAAA), which has crippled DNA- but not RNA-editing activity, further indicate that the ssDNA deamination activity of this enzyme is critical for triggering tumor development. These results are consistent with the recent literature showing that A3A expression can inflict DNA damage and APOBEC3 signature mutations in human cancer cell lines [36,37]. Taken together with our extensive observations in vivo, we propose that A3A be considered as a “master driver” of cancer that inflicts genome-wide DNA damage and a cacophony of mutational events that combine to trigger carcinogenesis and ultimately manifest as visible tumors.

An interesting observation shown in our original studies [24] and repeated here in several independent experiments is that A3A expression is lost during tumor development. We reason that this is a relatively early event because both A3A and A3A-E72A show robust staining by IHC 48–72 h post-hydrodynamic injection, and only wildtype A3A but not catalytic mutant A3A expression is lost by the earliest timepoints of visible tumor detection (6–8 weeks p.i.). Moreover, there is no apparent selective disadvantage to A3A-E72A expression in this system, such as immune rejection, as near-homogeneous staining is seen in fully regenerated livers 6 months p.i. Taken together with evidence for elevated DNA damage in A3A-expressing cells (g-H2AX staining here and ref. [24]) and high levels of SBS2 and SBS13 in resulting HCC lesions [24], we postulate that A3A causes a large number of DNA deamination events that either kill individual expressing hepatocytes or select against A3A itself while simultaneously creating the mutations required for liver carcinogenesis (as well as many passenger mutations). Sizable numbers of tumor genomic DNA sequences will be required to identify specific pathways and individual genes that contribute downstream of A3A to this overall carcinogenic process.

RNA editing has the potential to be as phenotypically profound as heritable mutations at the DNA level [18,38,39,40]. Moreover, a significant number of studies have linked epigenetic RNA-editing events to cancer phenotypes, including non-small cell lung cancer [41], multiple myeloma [42,43], and peripheral nerve sheath tumors [44,45,46]. Perhaps the most notable prior work linked Apobec1, which is a bona fide RNA-editing enzyme, to HCC [47]. Likewise, transgenic overexpression of the antibody gene deaminase AID, which has also been hypothesized to edit RNA, is capable of causing T cell lymphomagenesis in mice [48]. However, it subsequently became appreciated that, like A3A, Apobec1 and AID are potent ssDNA-editing enzymes [49,50,51,52]. Thus, the precise contributions of DNA versus RNA editing by polynucleotide deaminase family members in cancer have been difficult to tease apart until now. The results here with elevated DNA damage (g-H2AX staining) in human A3A expressing murine hepatocytes, a requirement for the catalytic glutamate E72 in tumor formation, and the additional dependency on DNA deamination activity indicated by the A3A separation-of-function quadruple mutant GAAA combine to demonstrate that DNA editing is the major activity required for tumor formation. The fact the A3A expression is lost in observed HCCs (but the mutations it inflicts are not) is further consistent with a heritable DNA-level, not a transient RNA-level, editing mechanism of carcinogenesis. Future studies will be needed to comprehensively characterize all DNA- and RNA-editing sites in this system and, importantly, to extend results from mice here to human malignancies impacted by APOBEC3 signature mutations.

## 4. Materials and Methods

### 4.1. Animal Care

Animals were maintained in specific pathogen-free facilities maintained by the Research Animals Resources facility at the University of Minnesota (Minneapolis, MN, USA) where they had free access to food and water and were kept on a 12 h light-dark cycle. All studies were performed in accordance with the recommendations in the Guide for the Care and Use of Laboratory Animals of the National Institutes of Health. The experiments performed with mice in this study were approved by the University of Minnesota Institutional Animal Care and Use Committee (protocol 2004-38049A).

### 4.2. Cell Culture and Reagents

The 293T and HeLa cells were purchased from ATCC and maintained in complete RPMI 1640 (Gibco, ThermoFisher Scientific, Waltham, MA, USA) with 10% heat-inactivated fetal bovine serum (FBS, Corning) and 1% antibiotic/antimycotic (Gibco, ThermoFisher Scientific, Waltham, MA, USA). Cells were cultured in a humidified atmosphere of 5% CO_2_ at 37 °C.

### 4.3. Cloning

A schematic of the plasmids used in Figure 1 is shown in Appendix A, the plasmids used in Figure 2A–C and Figure 3A–G are shown in Appendix A, and the constructs used in Figure 4A–H are shown in Appendix A. The construction of the plasmids in Appendix A was described previously [24]. Briefly, intron-containing human A3A or A3A-E72A cDNA sequences were ordered as gBlocks (IDT) and cloned into the pENTR entry vector (Invitrogen, ThermoFisher Scientific, Waltham, MA, USA) using NotI-NcoI. The final Gateway destination plasmid coexpressing Fah, GFP, and luciferase was combined with pENTR-A3 vectors using Gateway LR clonase mix (ThermoFisher Scientific, Waltham, MA, USA, catalog no. 11791-020) to generate the pT2/GD-Fah-A3A or pT2/GD-Fah-E72A delivery plasmids (pRH9777 and pRH9785, respectively). The transposon vectors expressing an shRNA against Trp53 or an shRNA control and validation in vivo have been described [28] (shp53, pRH5081; shCon, pRH9840; Appendix A).

The construction of the constructs shown in Appendix A is as follows: a base vector pcDNA-SB-Control (pRH10360) was constructed by PCR amplifying the left Sleeping Beauty IR/DR with forward primer 5′-GACTCAAGATCTGGCGAATTGGAGCTC and reverse primer 5′-CCAAGCAATTGAAAGGCACAGTCAAC and the right IR/RD with forward primer 5′-CGCGTGTATACGCTACCAAATACTAA and reverse primer 5′-CGCGTACATGTTCGACTCTAGCTAGA. The left was digested with BglII, the right was digested with BstZ17I and PciI (New England Biolabs, Ipswich, MA, USA). Vector pcDNA3.1 was digested with either BglII and NruI (left) or BstZ17I and PciI (right) (all purchased from New England Biolabs, Ipswich, MA, USA), and the appropriate products were ligated together. After sequencing each plasmid, the fragments were combined by digestion with SalI and AvrII (New England Biolabs, Ipswich, MA, USA) followed by ligation with the left and right IR/DRs into one plasmid with a central multiple cloning site preceded by a CMV promotor [36].

The A3A-Y132G-G188A-R189A-L190A (GAAA; pRH10511) quadruple mutant was generated by a series of site directed mutagenesis (SDM) reactions and ligations. Briefly, the Y132G and R189A mutations were made by 2 independent SDM reactions on a wildtype A3A sequence (pRH10502 and pRH10503, respectively). A Y132G-R189A double mutant was made by digesting each of the Y132G and R189A single mutant plasmids with EcoNI and PciI (New England Biolabs, Ipswich, MA, USA), gel purifying the products and ligating the appropriate fragments (pRH10504). A quadruple mutant Y132G-G188A-R189A-L190A (pRH10511) was generated by two rounds of SDM on the A3A-Y132G-R189A. PCR amplification was used to generate each cDNA fragment. The PCR inserts and vector were digested with NheI and NotI (New England Biolabs, Ipswich, MA, USA), gel purified, and the appropriate fragments were ligated, recovered in *E. coli*, and confirmed by DNA sequencing.

### 4.4. Liver Regeneration Experiments

*Fah*-deficient mice expressing SB11 (Fah^−/−^; Rosa26-SB11Tg/WT) were generated and maintained with drinking water containing 7.5 µg/mL NTBC (Sigma-Aldrich, St. Louis, MO, USA, product no. PHR1731) as described [30]. At 8–10 weeks of age, male and female mice were randomly enrolled into experimental groups and underwent hydrodynamic tail vein injections to integrate transposon vectors. Immediately before hydrodynamic delivery, mice were anesthetized by administering 25 µL anesthetic cocktail (8 mg/mL ketamine HCl, 0.1 mg/mL acepromazine maleate, and 0.01 mg/mL butorphanol tartrate) via intraperitoneal injection (i.p.). Each animal was injected with a total of 20 µg of transposon plasmid (PureLink HiPure Plasmid Filter Maxiprep kit; Invitrogen, ThermoFisher Scientific, Waltham, MA, USA) diluted in lactated Ringer’s solution (ThermoFisher Scientific, Waltham, MA, USA) to an injection volume equivalent to 10% of the body weight of the mouse (10% vol/body wt). Immediately following injection, animals were placed on normal drinking water to promote liver repopulation with Fah transgenic cells. Six months after injection, mice were euthanized and weighed, and liver tissues were harvested and weighed. Liver mass was recorded and all visible nodules (>2 mm in diameter) were counted and carefully isolated from neighboring non-tumorous tissue. Visible tumors were only detected in liver tissues and never at other sites despite the initial intravenous delivery of plasmid DNA.

### 4.5. Immunoblotting

Semi-confluent HeLa cells (Appendix A) or 293T cells (Figure 4B) were grown in a 6-well plate and were transfected with 2.5 µg of plasmid DNA in 7.5 µL of TransIT-LT1 (Mirus Bio, Madison, WI, USA, catalog no. MIR 2300) and 250 µL Opti^TM^-MEM reduced serum medium (Gibco, ThermoFisher Scientific, Waltham, MA, USA). The cells were resuspended in HED buffer (1 million cells/100 µL of HED buffer, 1× HED buffer: 25 mM Hepes, 5 mM EDTA, 10% glycerol, 1 mM DTT, and 1× protease inhibitor cocktail tablet (cOmplete; Roche, MilliporeSigma, Burlington, MA, USA, product no. 11836170001)), lysed by multiple freeze-thaw cycles, and then sonicated in the water bath for 20 min at 4 °C. Protein lysates from mouse livers (Figure 3C,G) were obtained by pushing the tissues through a 70 μm filter and resuspending the cells in PBS at 1 μL/mg of tissue. The cells were lysed by multiple freeze–thaw cycles, and then sonicated in the water bath for 20 min at 4 °C. Then, 2× RSB (1× RSB solution: 62.5 mM Tris-Cl, pH 6.8, 20% glycerol, 7.5% SDS, 5% 2-mercaptoethanol, and 250 mM DTT) was added to the lysate in a 1:1 ratio and samples were heated in a 98 °C water bath for 30 min. Proteins were separated by a 12.5% SDS-PAGE gel and transferred to polyvinylidene Immobilon-FL membranes (Millipore, Burlington, MA, USA). Membranes were blocked in blocking solution (5% milk + PBS supplemented with 0.1% Tween 20) and then incubated with primary antibody diluted in blocking solution. Primary immunoblotting antibodies were mouse α-tubulin (1:10,000 dilution, Sigma-Aldrich, St. Louis, MO, USA, product no. T5168), rabbit α-human A3A/B/G mAb (1:1000 dilution, 5210-87-13 [53]), or rabbit α-human A3A mAb (1:500 dilution, UMN-13). Secondary antibodies were diluted in blocking solution supplemented with 0.02% SDS. Secondary antibodies used for detection were goat α-mouse 680LT (LI-COR Biosciences, Lincoln, NE, USA, product no. 925–68020) and goat α-rabbit HRP (Cell Signaling Technology, Danvers, MA, USA, product no. 7074P2;). Proteins labeled with HRP-conjugated antibodies were detected with SuperSignal West Femto Maximum Sensitivity Substrate (ThermoFisher Scientific, Waltham, MA, USA, catalog no. 34095). Membranes were imaged with an Odyssey Classic scanner and Odyssey Fc imager (LI-COR Biosciences, Lincoln, NE, USA) or a ChemiDoc MP Imaging System (Bio-Rad, Hercules, CA, USA).

### 4.6. Deaminase Activity Assays

Semi-confluent HeLa cells (Appendix A) or 293T cells (Figure 4B) were grown in a 6-well plate and were transfected with 2.5 µg of plasmid DNA in 7.5 µL of TransIT-LT1 (Mirus Bio, Madison, WI, USA, catalog no. MIR 2300) and 250 µL Opti^TM^-MEM reduced serum medium (Gibco, ThermoFisher Scientific, Waltham, MA, USA). The cells were resuspended in HED buffer (1 million cells/100 µL of HED buffer, 1× HED buffer: 25 mM Hepes, 5 mM EDTA, 10% glycerol, 1 mM DTT, and 1× protease inhibitor cocktail tablet (cOmplete; Roche, MilliporeSigma, Burlington, MA, USA, product no. 11836170001)), lysed by multiple freeze–thaw cycles, and then sonicated for 20 min in a water bath and cleared by centrifugation. Protein concentration in lysates was quantified with a NanoDrop (ThermoFisher Scientific, Waltham, MA, USA). Then, 75 µg of protein lysate was incubated with 800 nM 5′-ATTATTATTATTCGAATGGATTTATTTATTTATTTATTTATTT-FAM oligo at 37 °C with 100 µg/mL RNase A and 0.1 U uracil DNA glycosylase (New England Biolabs, Ipswich, MA, USA) for 24 h. NaOH was added to a final concentration of 100 mM, and samples were heated to 98 °C for 10 min to cause breakage of abasic sites caused by deamination and subsequent uracil removal. Samples were separated by 15% TBE-Urea PAGE to resolve product and imaged on a Typhoon FLA 7000 biomolecular imager (GE Healthcare Life Sciences, Chicago, IL, USA). Band volume intensities (ratio of substrate to product) were quantified by densiometry with ImageQuant TL software (GE Healthcare Life Sciences, Chicago, IL, USA).

### 4.7. RT-qPCR for P53 Knockdown

RNA was extracted from livers with RNeasy Mini Kit (Qiagen). cDNA was prepared using Transcriptor reverse transcription (Roche, MilliporeSigma, Burlington, MA, USA, product no. 3531317001) with random hexamer priming. Relative transcript levels were measured by quantitative PCR with SsoFast EvaGreen Supermix (Bio-Rad, Hercules, CA, USA, product no. 1725201) on a LightCycler 480 instrument (Roche, MilliporeSigma, Burlington, MA, USA) and the following primers: Tbp: mouse_Tbp_Forw 5′-GGGGAGCTGTGATGTGAAGT, mouse_Tbp_Rev 5′-CCAGGAAATAATTCTGGCTCA; Trp53: mouse_Trp53_Forw 5′-AAGTCACAGCACATGACGGA, mouse Trp53_Rev 5′-CCGGATAAGATGCTGGGGAG.

### 4.8. RNA-Editing Experiments

The 293T cells were transfected as above with A3A, E72A, GAAA, or control constructs (pRH10364, pRH10365, pRH10511, and pRH 10360, respectively; Appendix A) and, 48 h post-transfection, RNA was extracted from cells with a RNeasy Mini Kit (Qiagen, Hilden, Germany, catalog no. 74106). cDNA was prepared using Transcriptor reverse transcription (Roche, MilliporeSigma, Burlington, MA, USA, product no. 3531317001) with random hexamer priming. PCR amplification of the hairpin region of the *SDHB* transcript was performed using forward primer 5′-GGTCCTCAGTGGATGTAGGC and reverse primer 5′-GTCAAAGTAGAGTCAACTTCAT. The PCR cycle was performed as the following: initial denaturation 98 °C for 3 min, then denaturation at 98 °C for 10 s, annealing at 65 °C for 30 s, and extension at 72 °C for 45 s for 30 cycles, completed with a final extension at 72 °C for 7 min. PCR cleanup was carried out using GeneJet PCR Purification Kit (ThermoFisher Scientific, Waltham, MA, USA, product no. K0701). Purified PCR product (1 ng/µL), primer (20 pmol/µL) were brought to 15 µL with H_2_O and submitted for Sanger sequencing using reverse primer 5′-GTCAAAGTAGAGTCAACTTCAT. RNA editing of *SDHB* C136-to-U was quantified using EditR [54].

### 4.9. Histology

Animal livers were isolated and fixed in 10% neutral formalin. Hematoxylin and eosin (H&E) staining of the FFPE specimens was performed as follows. Tissues were sectioned at 4 µm, mounted on positively charged adhesive slides, and allowed to air-dry for at least 24 h before staining. To deparaffinize and rehydrate the samples, slides were baked in a 60–62 °C oven for 20 min, washed three times with xylene for 5 min each, soaked in graded alcohols (100% × 2, 95%, and 80% for 3 min each), and then rinsed in running water for 5 min. The tissues were stained with hematoxylin for 5 min and rinsed in running water for 30 s, followed by two dips in acid solution and 30–90 s in ammonia water (bluing solution). After a 10 min water rinse, the slides were transferred in 80% ethanol for 1 min, counterstained with eosin for 1 min, dehydrated in graded alcohols and xylene, and coverslipped with Cytoseal (ThermoFisher Scientific). The H&E-stained slides were subsequently scanned at 40× magnification and visualized using the Aperio ScanScope XT system (Leica Biosystems, Wetzlar, Germany) as described [53].

### 4.10. Immunohistochemistry

IHC was performed as described [53,55,56,57]. First, 4 µm-thick sections of FFPE liver tissues were mounted on positively charged, adhesive slides and allowed to air-dry for at least 24 h. To deparaffinize and rehydrate the samples, slides were baked in a 65 °C oven for 20 min, washed three times with CitriSolv (Decon Labs, King of Prussia, PA, catalog no. 1601) or xylene for 5 min each, soaked in graded alcohols (100% × 2, 95%, and 80% for 3 min/each), and then rinsed in running water for at least 5 min. Epitope retrieval was performed using Reveal Decloaker (BioCare Medical, Pacheco, CA, USA, product no. RV1000M) in a steamer for 35 min, followed by a 20 min “cool-down” period. Then, slides were rinsed with running tap water for 5 min and transferred to Tris-buffered saline with 0.1% Tween 20 (TBST) for 5 min. Endogenous peroxidase activity was quenched by placing the slides in 3% hydrogen peroxide in TBST for 10 min at room temperature, followed by a 5 min rinse under running water. To block nonspecific binding of primary antibody, sections were soaked in Rodent Block M (BioCare Medical, Pacheco, CA, USA, product no. RBM961) for 15 min at room temperature. After blocking, sections of each specimen were incubated overnight at 4 °C with a rabbit α-human A3A/B/G mAb (5210-87-13 [53]) diluted 1:350 or a rabbit α-human A3A mAb (UMN-13) diluted 1:6000 in 10% Rodent Block M in TBST. The rabbit α-human A3A mAb UMN-13 was generated by immunizing rabbits (performed by Labcorp, Burlington, NC, USA) with a peptide encompassing A3A residues 1–17 (MEASPASGPRHLMDPHI). Hybridomas were made using splenic B lymphocytes (Abcam, Cambridge, UK) and were screened by ELISA, immunoblotting [36], IF microscopy, and IHC as described above. Following overnight incubation with the primary antibodies, sections were rinsed in TBST for 5 min and completely covered with anti-rabbit poly-HRP-IgG (Novolink Polymer, Leica Biosystems, Wetzlar, Germany, product no. RE7260-K) for 30 min at room temperature. The reaction product was developed using the Novolink DAB substrate kit (Leica Biosystems, Wetzlar, Germany, product no. RE7230-K) at room temperature for 3 min, rinsed in tap water for 5 min, counterstained in Mayer’s hematoxylin solution (Electron Microscopy Sciences, Hatfield, PA, USA, product no. 26252–01) for 5 min, dehydrated in graded alcohols and CitriSolv, and coverslipped using Cytoseal. A3A immunoreactivity was visualized using the Aperio ScanScope XT platform (Leica Biosystems, Wetzlar, Germany).

### 4.11. Immunofluorescence Microscopy

Semi-confluent transfected HeLa cells were grown on 96-well glass-bottom black plates (Ibidi, Fitchburg, WI, USA). Cells were fixed in 4% paraformaldehyde in PBS for 15 min at room temperature. Cells were washed with cold PBS for 5 min and then permeabilized in 0.2% Triton X-100 in PBS for 10 min at 4 °C. Cells were washed once with cold PBS for 5 min. Cells were incubated in a blocking solution of 5% normal goat serum (Corning, Somerville, MA, USA) and 4% bovine serum albumin (ThermoFisher Scientific, Waltham, MA, USA) in PBS for 1 h at room temperature on a rocker. Primary antibody in blocking solution was added, and the cells were incubated overnight at 4 °C. Cells were washed with PBS 3× for 5 min at room temperature. Cells were incubated with secondary antibodies at a concentration of 1:1000 in blocking solution for 2 h in the dark at room temperature with gentle rocking. Cells were washed with PBS 3× for 5 min at room temperature before being imaged on a Nikon Inverted TI-E Deconvolution microscope (Nikon, Tokyo, Japan). Primary antibody was rabbit α-γ-H2AX mAb (Cell Signaling Technology; 1:200 dilution, catalog no. 9718S). Secondary antibody was goat anti-rabbit IgG (H + L) Highly Cross-Adsorbed Secondary Antibody, Alexa Fluor™ 647 (Invitrogen, ThermoFisher Scientific, Waltham, MA, USA, catalog no. A-21245). Cells were counterstained with Hoechst 33342 Ready Flow™ Reagent (Invitrogen, ThermoFisher Scientific, Waltham, MA, USA, catalog no. R37165) to visualize DNA content.

### 4.12. Real-Time DNA-Editing Assays

Real-time DNA-editing assays were performed as described [35,58]. Briefly, semi-confluent 293T cells in a 24-well plate format were transfected 25 min at RT with 100 ng gRNA, 100 ng reporter, 200 ng Cas9n-UGI-NLS, and 30 ng of either A3A, E72A, GAAA or control vectors (pRH10364, pRH10365, pRH10511, and pRH10360, respectively; Appendix A), 43 µL of serum-free RPMI 1640 (Gibco, ThermoFisher Scientific, Waltham, MA, USA), and 1.3 µL of TransIT-LT1 (Mirus Bio, Madison, WI, USA, catalog no. MIR 2300). Then, 48 h post-transfection, cells were imaged and analyzed by Cytation1 Multi-Mode Reader (BioTek, Santa Clara, CA, USA) configured with Texas Red (586/647) and GFP (469/525) light cubes using a 10× objective. All mCherry positive cells were first identified as the primary mask using size exclusion and a Texas Red fluorescence threshold, and eGFP-positive cells were then identified within the primary mask using a GFP fluorescence threshold. The ratio of eGFP positive cells to mCherry positive cells was calculated to indicate the editing percentage (Figure 4C).

## Figures and Tables

**Figure 1 ijms-24-09305-f001:**
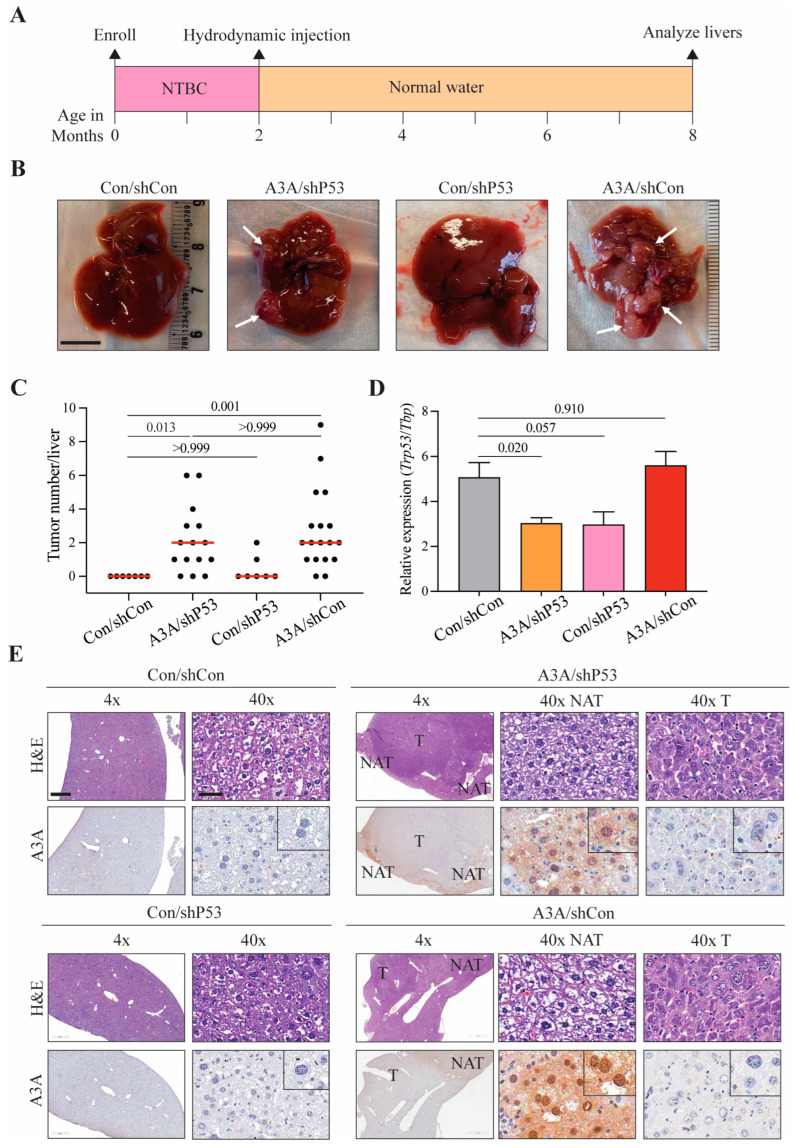
Human A3A induces HCC in vivo. (**A**) Experimental timeline with hydrodynamic injection and NTBC withdraw at 2 months, followed by liver analyses 6 months later. (**B**) Gross anatomy analysis of representative livers from each cohort. White arrows indicate tumorous nodules. Scale bar, 1 cm. Control (Con); non-shRNA control (shCon); shRNA P53 knockdown (shP53). (**C**) A plot indicating the number of macroscopic liver lesions. Red line indicates median tumor number (*n* = 7–16 animals per condition). *p* values calculated by Kruskal–Wallis non-parametric test with Dunn’s adjustment for multiple comparisons. (**D**) Qualitative RT-qPCR analysis of *P53* mRNA expression relative to *TBP*. Mean + SEM plotted. *p* values calculated by Brown–Forsythe and Welch one-way ANOVA with Dunnett’s adjustment for multiple comparisons. (**E**) Representative H&E and A3A (5210-87-13 mAb) IHC photomicrographs of liver tissues with and without lesions. Scale bar, 500 µm or 60 µm for 4× and 40× magnification, respectively, and inset images are magnified four-fold; tumor (T); normal adjacent-to-tumor (NAT).

**Figure 2 ijms-24-09305-f002:**
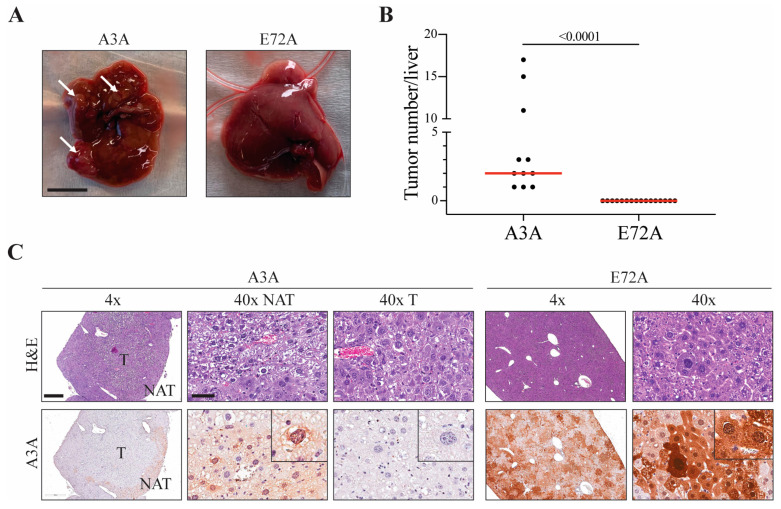
The catalytic glutamate of A3A is required for tumorigenesis. (**A**) Gross anatomy analysis of representative livers from the A3A and E72A conditions. White arrows indicate tumorous nodules. Scale bar, 1 cm. (**B**) Number of macroscopically detected lesions on liver surfaces. Red line indicates median tumor number (*n* = 11–16 animals per condition). Mann–Whitney non-parametric test, *p* < 0.001. (**C**) H&E and IHC (UMN-13 mAb) staining for A3A in representative livers and tumors (left panels A3A; right panels E72A). Scale bar, 500 or 60 µm for 4× and 40× magnification, respectively, and inset images are magnified four-fold; tumor (T); normal adjacent-to-tumor (NAT).

**Figure 3 ijms-24-09305-f003:**
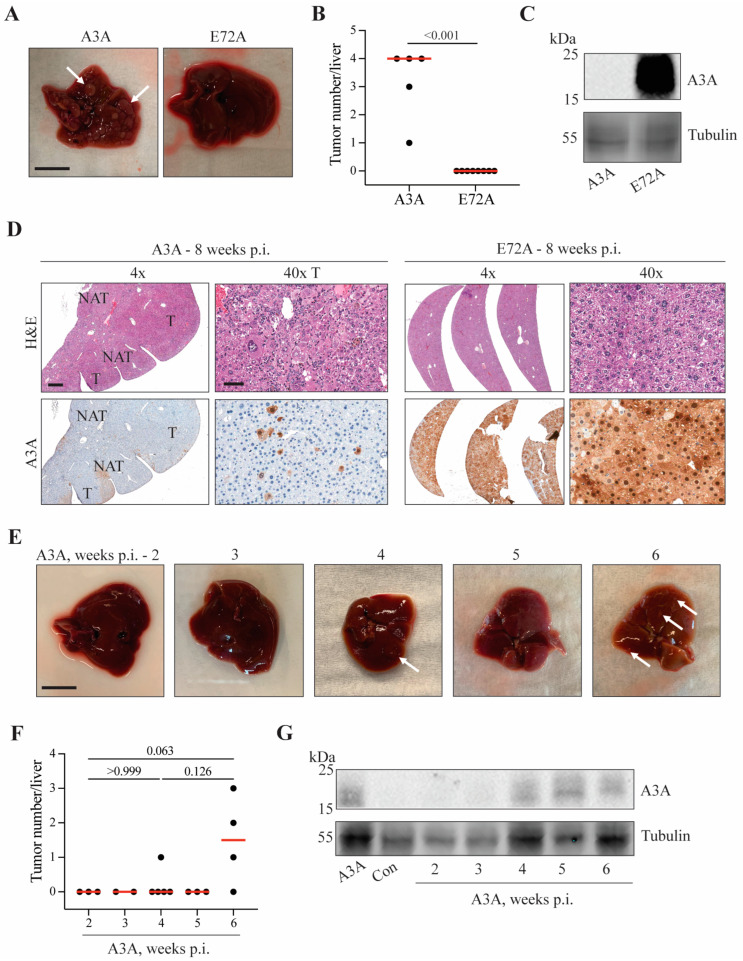
A3A can induce tumorigenesis in Fah livers within 6 weeks p.i. (**A**) Gross anatomy of representative livers harvested at 8 weeks p.i. White arrows indicate tumorous nodules. Scale bar, 1 cm. (**B**) Number of macroscopic tumors observed on livers harvested at 8 weeks. Red line indicates median tumor number (*n* = 5 animals per condition). *p* < 0.001 by Mann–Whitney non-parametric test. (**C**) Immunoblot for A3A expression (UMN-13 mAb) in livers harvested at 8 weeks. (**D**) H&E and IHC (UMN-13 mAb) of livers harvested 8 weeks p.i. Scale bar, 500 or 60 µm for 4× and 40× magnification, respectively. Tumor (T); normal adjacent-to-tumor (NAT). (**E**) Gross anatomy of representative livers harvested 2–6 weeks p.i. from the A3A cohort. White arrows indicate tumorous nodules. Scale bar, 1 cm. (**F**) Number of macroscopic liver lesions in animals 2–6 weeks p.i. Red line indicates median tumor number (*n* = 2–5 animals per condition). *p* values calculated by Kruskal–Wallis non-parametric test with Dunn’s adjustment for the three primary comparisons. (**G**) Immunoblot for A3A protein expression (UMN-13 mAb) in livers harvested 2–6 weeks p.i.

**Figure 4 ijms-24-09305-f004:**
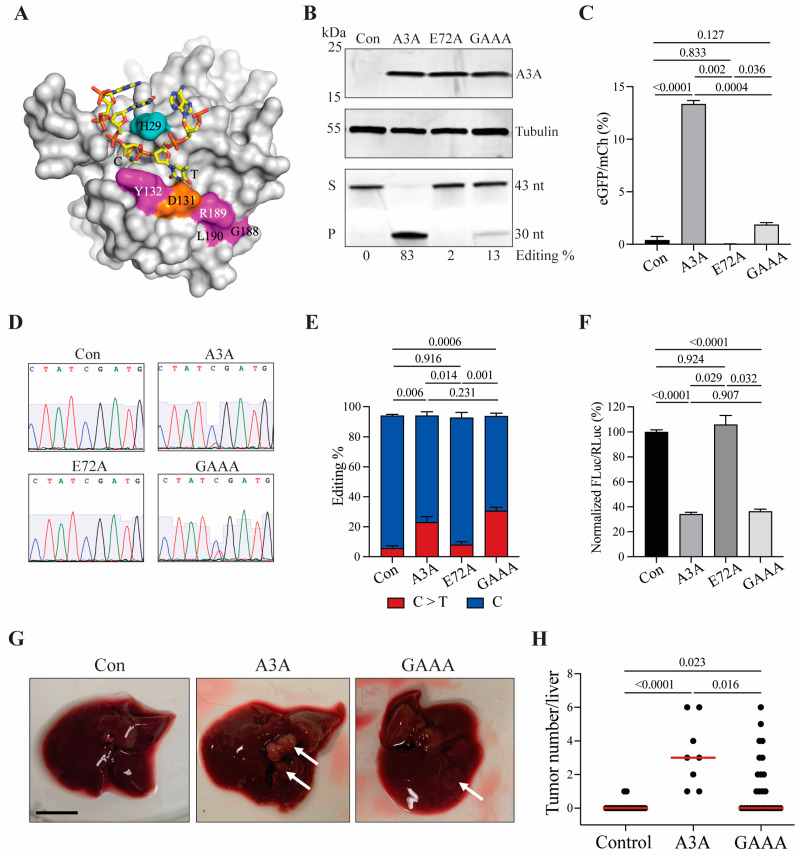
A3A-induced tumorigenesis occurs through a DNA deamination mechanism. (**A**) Structure of A3A bound to single-stranded DNA (orange and yellow). Key A3A residues are highlighted in color and are numbered (Y132, G188, R189, and L190). (**B**) Immunoblot (upper panels; UMN-13 mAb) and DNA deaminase activity assay (lower panel) of whole cell lysates of 293T cells transfected with constructs containing A3A, E72A, GAAA, or control. Control (Con); substrate (S); product (P); nucleotides (nt). DNA deaminase activity quantification of band volume intensities (ratio of substrate to product) by densitometry reported below as Editing %. (**C**) Quantification of real-time DNA editing using a cell-based gain-of-signal fluorescent reporter (AMBER [35]). Mean + SEM plotted. *p* values calculated by one-way ANOVA with Tukey’s adjustment for multiple comparisons. (**D**) Representative chromatograms generated by Sanger sequencing. Sequences are of the A3A-targeted cytosine at position C136 within the *SDHB* transcript, reverse-transcribed and PCR-amplified prior to sequencing. (**E**) Quantification of C-to-U editing of cytosine at position C136 within the *SDHB* transcript. Mean + SEM plotted. *p* values calculated by one-way ANOVA with Tukey’s adjustment for multiple comparisons. (**F**) Quantification of RNA editing using a real-time luciferase inactivation assay. Mean + SEM plotted. *p* values calculated by one-way ANOVA with Tukey’s adjustment for multiple comparisons. (**G**) Gross anatomy of representative livers harvested at 6 months p.i. from control, A3A, or GAAA injected animals. White arrows indicate macroscopic tumorous nodules. Scale bar, 1 cm. (**H**) Number of macroscopic lesions observed on livers. Red line indicates median tumor number (*n* = 8–27 animals per condition). *p* values calculated by Kruskal–Wallis non-parametric test with Dunn’s adjustment for multiple comparisons.

## Data Availability

All data are available in the main manuscript or Appendix A and additional data and reagents can be obtained by written request to rsh@uthscsa.edu.

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
