# Peer review of "DNA Deamination Is Required for Human APOBEC3A-Driven Hepatocellular Carcinoma In Vivo"

_ijms, 2023, doi:10.3390/ijms24119305_

Round 1

Reviewer 1 Report

This manuscript from Naumann et al follows recent work from the same group showing that APOBEC3A overexpression can promote tumorigenesis across multiple mouse cancer models. Here, the authors expand on the original work by showing that APOBEC3A deaminase activity is required for enhanced tumorigenesis and that p53 depletion and APOBEC3A-dependent RNA editing are dispensable for enhanced tumorigenesis. Tracking tumor development across time shows how tumors can develop in two months. In particular, the demonstration here that APOBEC3A catalytic activity is essential for tumor formation is a critical result that suggests against deaminase-independent functions (i.e. potential impediment to replication forks). New findings in the field indicate that APOBEC3A possesses RNA editing activity as well as DNA editing activity. This presents an interesting open question of how APOBEC3A may be driving tumor formation in the Fah model: ssDNA vs RNA editing activity. By using a clever separation of function mutant that diminishes APOBEC3A activity against ssDNA in a targeted manner that preserves RNA editing activity, the authors are able to link tumor promoting capabilities to APOBEC3 ssDNA editing function. Overall, the manuscript consists of rigorous experimentation and tackles important open questions in the field. I only have minor suggestions for discussion and additional context.

Figure 1: I am surprised that p53 depletion has such a modest effect on liver tumor formation. Perhaps the authors can comment or place this result in additional context. For example, is this due to the relatively early time points being investigated? 

Figure 1: Related to the point above: Does p53 status play any role in selecting for loss of APOBEC3A expression at early time points? I am not sure that any additional experimentation is necessary to address these points but additional commentary would be helpful.

Figure 4: In order to further characterize the APOBEC3A GAAA mutant it may be informative to look for APOBEC3A loss from tumors that develop in these animals (at a later time point than is already addressed in the manuscript). Residual tumor-promoting activity may result from weak, but persistent APOBEC3A deaminase activity. Failure to clear APOBEC3A expression from developed tumors would also be consistent that APOBEC3A DNA deaminase activity is being significantly affected by the GAAA mutant and thus loosening the demand for APOBEC3A loss. In the future DNA sequencing and comparison of mutations in the wild-type and GAAA tumors may prove informative.

Minor point: No relevant reference to a recent report indicating that APOBEC3A is a major driver of mutations in human cancer cells (35859169). This prior study helps provide additional rationale for the experiments described in this current manuscript.

Minor point: I suspect that line 341 contains a typographical error: ssDNA vs RNA.

Reviewer 2 Report

In this study, Naumann et al demonstrated that ectopic expression of wild-type APOBEC3A(A3A) was sufficient for carcinogenesis in Fah-mutant liver, even when a tumor suppressor, TP53 was physiologically expressed. They further compared tumorigenecity of wid-type A3A with those of functional mutants, and proposed that its DNA, but not RNA editing activity was required for the tumor formation.

The study is well organized and the conclusion is clear.

I have a single concern regarding the conclusion from Fig.4, where  the authors concluded that  

"A3A-catalyzed DNA (and not RNA) editing activity is responsible for the tumor phenotypes described here using the Fah HCC model system (Lines 303-305)."

"The results here with the separation-of-function quadruple mutant GAAA strongly indicate that DNA editing is the major activity required for tumor formation  (Lines 343-344)."

Fig.4H indicated significantly increased tumor numbers in GAAA mutants, compared to the Control  (p=0.023) They should not completely deny contribution of RNA editing activity to the tumorigensis.

Reviewer 3 Report

The well-written and significant manuscript presents an excellently executed study of human APOBEC3A (A3A) effects on tumor formation in the murine Fah liver complementation and regeneration model. It provides evidence that DNA and not RNA deamination is responsible for cancers.

It is well-known that overexpression of editing deaminases in mice causes an increase in cancer occurrence (starting from 2003, J Exp Med. 2003;197(9):1173-81), but there is some debate what is the cause, RNA or DNA deamination. The problem looks somewhat wiredrawn because APOBECs are mutagenic, and mutagens cause cancer, bringing us fifty years back to the era of Bruce Ames's studies of the correlation between mutagens and carcinogens. Still, in theory, RNA editing could be a bottleneck in cancer development. The current work addresses the controversy with the tool of strategic A3A mutants, deficient either in DNA and RNA cytosine deamination (A3A-E72A) or decreasing only DNA deamination (A3A-GAAA mutant). As designed, this would have allowed us to see that DNA deamination causes cancer. However, the A3A-GAAA mutant has just a decrease, not a lack of DNA deamination activity, and tumors are still induced by this mutant, albeit with lower frequency, which leaves room for skepticism. There might be RNA targets that the A3A-GAAA mutant cannot access, and this will cause a decrease. For example, APOBEC1's primary role is the deamination of only one mRNA, though it is a good DNA mutagen. The relation between APOBEC abundance and activity is non-linear (Mol Cell Biol. 2018;39(1):e00238-18), so the decrease of tumor formation postulated by the authors (though not very clearly seen in Fig. 4H) may be a result of lower DNA deamination or lack of deamination of unknown, specific RNA targets. Despite the imperfection, the study is significant and impressive. The immunohistochemistry results are inspiring. We highlight the finding that A3A is not detected in the already formed tumors that A3As, a confirmation of earlier studies. Therefore, active A3A is lost during early tumor development. Discussing the previous data on high APOBEC abundance in human cancerous tissues and cell lines would be interesting. Otherwise, the future directions (lines 345-346) look generic.

Minor comments.

Line 43. It would be good to mention the POLE's role in ultramutated tumors; the current list looks outdated.

Lines 303-304. The conclusion that the results “indicate…” looks like uncertainty.

In Figures 2B and 4H, it is impossible to see the number of animals analyzed. At first glance, the tumorigenic activity of A3A-GAAN looks similar to A3A; the median bar hinders symbols livers with low tumor burden.
